# Predictive Modeling of Injury Risk Based on Body Composition and Selected Physical Fitness Tests for Elite Football Players

**DOI:** 10.3390/jcm11164923

**Published:** 2022-08-22

**Authors:** Francisco Martins, Krzysztof Przednowek, Cíntia França, Helder Lopes, Marcelo de Maio Nascimento, Hugo Sarmento, Adilson Marques, Andreas Ihle, Ricardo Henriques, Élvio Rúbio Gouveia

**Affiliations:** 1Department of Physical Education and Sport, University of Madeira, 9020-105 Funchal, Portugal; 2Laboratory of Robotics and Engineering Systems, Interactive Technologies Institute, 9020-105 Funchal, Portugal; 3Institute of Physical Culture Sciences, Medical College, University of Rzeszów, 35-959 Rzeszów, Poland; 4Department of Physical Education, Federal University of Vale do São Francisco, Petrolina 56304-917, Brazil; 5University of Coimbra, Research Unit for Sport and Physical Activity (CIDAF), Faculty of Sport Sciences and Physical Education, 3004-504 Coimbra, Portugal; 6CIPER, Faculty of Human Kinetics, University of Lisbon, 1495-751 Lisbon, Portugal; 7ISAMB, Faculty of Medicine, University of Lisbon, 1649-020 Lisbon, Portugal; 8Department of Psychology, University of Geneva, 1205 Geneva, Switzerland; 9Center for the Interdisciplinary Study of Gerontology and Vulnerability, University of Geneva, 1205 Geneva, Switzerland; 10Swiss National Centre of Competence in Research LIVES—Overcoming Vulnerability: Life Course Perspectives, 1015 Lausanne, Switzerland; 11Marítimo da Madeira—Futebol, SAD, 9020-208 Funchal, Portugal

**Keywords:** sports injuries, machine learning, injury prediction, sports monitorization, elite football

## Abstract

Injuries are one of the most significant issues for elite football players. Consequently, elite football clubs have been consistently interested in having practical, interpretable, and usable models as decision-making support for technical staff. This study aimed to analyze predictive modeling of injury risk based on body composition variables and selected physical fitness tests for elite football players through a sports season. The sample comprised 36 male elite football players who competed in the First Portuguese Soccer League in the 2020/2021 season. The models were calculated based on 22 independent variables that included players’ information, body composition, physical fitness, and one dependent variable, the number of injuries per season. In the net elastic analysis, the variables that best predicted injury risk were sectorial positions (defensive and forward), body height, sit-and-reach performance, 1 min number of push-ups, handgrip strength, and 35 m linear speed. This study considered multiple-input single-output regression-type models. The analysis showed that the most accurate model presented in this work generates an error of RMSE = 0.591. Our approach opens a novel perspective for injury prevention and training monitorization. Nevertheless, more studies are needed to identify risk factors associated with injury prediction in elite soccer players, as this is a rising topic that requires several analyses performed in different contexts.

## 1. Introduction

Injuries are one of the most significant hampering issues for elite football players [1]. Football is known for its fast-paced and powerful actions [2,3], which might contribute to players’ increased risk of injuries [4]. Due to their effects on individuals’ mental states and overall teams’ performances, elite players’ injuries significantly impact the sports business [5,6]. Consequently, elite football clubs have been consistently interested in having practical, interpretable, and usable models as decision-making support for coaches and their technical staff members [7].

From the clinical standpoint, the literature describes the lower limbs as the most affected body zone by sports injuries [4,8,9,10,11,12,13,14], particularly for muscle injuries in the thigh area, the quadriceps, and the groin [4,10,15,16]. Since injuries in professional soccer are an increasingly problem, it is crucial that the work done in training sessions reflects the demands of competition, aiming at the development of athletes’ performance, which includes injury prevention [17,18,19].

Machine learning or statistical learning methods are currently tools that can significantly support decision-making in various aspects of the training process. For instance, it has been reported in the literature that some models can optimize training loads [20], which reinforces the applicability of machine learning in improving injury prediction [21,22].

Researchers, managers, and coaches are becoming increasingly involved in injury forecasting, using regular data collection that will allow them to act consciously and intervene on time on this global issue [23]. An investigation conducted over 18 years showed that the total injury rate in practice and competition has dropped during the past years [24]. Although the cause leading to this decrease is still unknown, one potential explanation for this decrease may be related to the effectiveness of injury prevention. If so, it is likely that the motivation of the medical staff at elite football teams is increasing, in terms of implementing and overseeing preventive injury programs [24].

Machine learning offers a modern statistical method that uses algorithms mainly created to deal with unbalanced data sets and enable the modeling of interactions between a large number of variables [25]. In the football context, machine learning has been used in injury prediction, physical performance prediction, training load and monitoring, players’ career trajectories, clubs’ performance, and match attendance [26].

There has been some research done on elite-football-injury prediction up to this point [23,25,27,28,29,30,31]. In 2019, 96 male elite football players participated in a study throughout a season, with hamstring-strain injuries being the primary anticipated consequence. In that study, the prediction model showed moderate to high accuracy for identifying players at risk of hamstring-strain injuries during pre-season testing [31]. Another example involved 26 elite football players participating in year-long research to forecast non-contact injuries. The authors reported that machine learning was far more accurate than baselines and modern injury-risk-estimating approaches, detecting roughly 80% of injuries with about 50% accuracy [23]. In another study conducted with 132 male elite football and handball players, the prediction model accurately identified elite players at risk of developing muscular injuries [25].

Two types of variables are highlighted in the previous research on predictive modeling of injury risk [30]. The first block of predictor variables is modifiable variables, i.e., training loads or physiological and physical fitness tests. The second type is non-modifiable variables, including demographic variables, anthropometric parameters, and injury histories. Indeed, body composition and physical fitness tests are the most commonly assessed by sports staff given their close relationship with game performance and players’ health. Moreover, evaluating and monitoring players’ characteristics during the season provides valuable information to understand better players’ behavioral changes and support coaches’ decision-making in the training and match process. In the sports injury literature, most of the investigation conducted aimed to assess one specific variable at a time to predict injury risk. However, this approach limits the correlation of injury risk and a global interpretation of players’ performance in professional football [23]. Therefore, this study aimed to analyze predictive modeling of injury risk based on body composition variables and selected physical fitness tests for elite football players across a sports season.

## 2. Materials and Methods

### 2.1. Participants

Thirty-six players from a professional football team participated in this study. This team competed in the First Portuguese League during the 2020/2021 season.

A description of the variables together with the basic statistics (M—mean value, SD—standard deviation) is given in Table 1. The models were calculated based on 22 independent variables (x_1_–x_22_) and one dependent variable (y). Independent variables include players’ information (sectorial position, age, experience, and number of previous injuries), anthropometric parameters with body composition, and components of physical fitness (flexibility, general strength, explosive strength, speed, agility, and aerobic endurance). The dependent variable is the number of injuries per season. The predictive analysis did not use the data of all athletes. Twenty-four players’ data were used. This was due to the fact that some of the athletes were noted to have missing data related to not taking certain physical fitness tests.

All procedures applied were approved by the Ethics Committee of the Faculty of Human Kinetics, CEIFMH No. 34/2021. The investigation was conducted following the Declaration of Helsinki, and informed consent was obtained from all participants.

### 2.2. Body-Composition Assessment

Body-composition variables were assessed using hand-to-foot bioelectrical impedance analysis (InBody 770, Cerritos, CA, USA). Height was measured to the nearest 0.1 cm using a stadiometer (SECA 213, Hamburg, Germany). The measurements occurred in the early morning, with participants fasting and wearing only their underwear. During the assessment, participants were barefoot, standing with both arms 45° apart from the trunk, with both feet bare on the spots of the platform. A total of 26 evaluations of body composition were considered during the season. Body mass, total body water (TBW), body fat mass (BFM), and fat-free mass (FFM) were retained for analysis.

### 2.3. Physical Fitness Assessment

The sit-and-reach bilateral test was used to evaluate flexibility measurement. A box (32.4 cm high and 53.3 cm long) with a 23 cm heel line mark was used. The participants sat barefoot in front of the box, with both knees fully extended and heels against the box. The research team held one hand lightly against each participant’s knees to ensure complete leg extension. Then, participants placed their hands on top of each other, palms down, and slowly bent forward along the measuring scale. The forward-hold position was repeated twice. The third and final forward stretch was held for three seconds, and the score was recorded to the nearest 0.1 cm.

The push-ups test protocol consisted in performing the highest number of push-ups in one minute, respecting the success criteria judged by the evaluator. The participants started the test in the down position to get correct hand placement and then assumed the up position, from which they did the maximum number of push-ups possible. No cadence was used, although participants were encouraged to execute push-ups with good form but fast enough to obtain the best possible score in a minute. The evaluator independently counted the number of push-ups correctly executed.

The handgrip protocol consisted of three alternated data collection trials for each arm, performed using a hand dynamometer (Jamar Plus+, Chicago, IL, USA). Participants were instructed to hold a dynamometer in one hand, laterally to the trunk with the elbow at a 90° position [32]. From this position, participants were instructed to squeeze as hard as possible, progressively and continuously squeezing the hand dynamometer for about two seconds. The dynamometer could not contact the participant’s body; otherwise, the trial was repeated. The best score of the three trials was retained for analysis. 

The countermovement jump (CMJ) and the squat jump (SJ) were used to assess lower-body explosive strength [33]. Both protocols included four data collection trials and were performed using the Optojump Next (Microgate, Bolzano, Italy) system of analysis and measurement. In both tests, participants were encouraged to jump to their maximum height. Before data collection, three experimental trials were performed by each participant to ensure correct execution. For the CMJ, participants began in a tall standing position, with feet placed hip-width to shoulder-width apart. Then, participants dropped into the countermovement position to a self-selected depth, followed by a maximal-effort vertical jump. Hands remained on the hips for the entire movement to eliminate any influence of arm swing. If the hands were removed from the hips at any point, or excessive knee flexion was exhibited during the countermovement, the trial was repeated. The participants reset to the starting position after each jump. The SJ protocol testing began with the participant in a squat position at a self-selected depth of approximately 90° of knee flexion, holding this position for the researchers’ count of three before jumping. If a dipping movement of the hips was evident, then the trial was repeated. The participants reset to the starting position after each jump.

Linear speed was assessed with maximal sprints at 5, 10, and 35 m, starting from a stationary position. Sprint time was recorded using Witty-Gate photocells (Microgate, Bolzano, Italy). Participants were allowed two trials for each sprinting distance, and the best time was used for analysis.

A yoyo intermittent recovery test was applied to evaluate the athlete’s maximum oxygen uptake under repeated high-intensity aerobic exercise [34,35]. The test consists of a 2 × 20 m shuttle run at increasing speeds, interspersed with 10 s of active recovery, controlled by audio signals. The test terminated when the subject was no longer able to maintain the required speed. The total distance and VO2 maximum record were used as results [36]. The results used were based on the athletes’ performance in the yoyo test, which is an indirect method of measuring such variables.

All tests were performed on the same day within a 4 h period in the morning (8 a.m.–12 p.m.). They were conducted by trained staff from the research team, who were familiar with each protocol. All protocols were followed with the utmost rigor, and the organization of the sequence of physical tests was designed to reduce the fatigue factor throughout all tests.

### 2.4. Injury Report

This study followed the Union of European Football Association (UEFA)’s recommendations for epidemiological investigations. An injury was defined as an event during a scheduled training session or match, resulting in an absence from the next training session or match [37]. Regarding the variables under analysis, the type, zone, and specific location of the injury are complementary variables that identify the part of the body that suffered structural and/or functional changes. The mechanism of injury is intended to understand if the injury was traumatic or if it was contracted by overload. The severity of the injury considers the period, in days, from the athlete’s stoppage until resuming field work with the consent of the clinical department. Finally, an injury was marked as recurrent when a player was injured in the same place and type where they were previously affected by an injury. Injury records during the season, including in training and competitive moments, were made daily by the clinical department.

### 2.5. Predictive Modeling

In this analysis, multiple-input single-output models for prediction were used. The output of the model is a continuous variable and represents the number of occurrences of potential injuries. Therefore, we consider regression-type models, not classifiers. Classic regression models (OLS), shrinkage regression, and stepwise regression were used in the models’ calculations. All predictive models were calculated using R Software version 4.2.0 (R Foundation for Statistical Computing, Vienna, Austria, 2022). The implemented methods included:The ordinary least squares regression (OLS) used a popular least-squares method, in which weights are calculated by minimizing the sum of the squared errors.The Ridge model was calculated using the criterion of performance, which includes a penalty for increased weights. Parameter *λ* decides the size of the penalty: the greater the value of *λ* is, the bigger the penalty. The value of lambda can vary from 0 to infinity [38].Lasso regression is the model where the mechanism facilitates assigning a penalty to variables, and, in this way, they are eliminated from equations. In Lasso regression [39], the parameter *s* (penalty) is used to optimize the model.Elastic net (ENET) [40] combines the features of ridge and LASSO regressions. The performance criterion is the so-called naive elastic net. To minimize the criterion, the LARS-EN algorithm was suggested [40], which is based on the LARS algorithm for LASSO regression. In elastic net regression, we have two parameters, penalty *s* and *λ*.Stepwise Forward Regression has a forward selection procedure (FS), which begins with an equation that contains only a free expression. The first variable in the equation is the one that has the highest correlation with the output variable. If the coefficient of regression of the variable differs significantly from zero, the variable remains in the equation and another variable is added. The second variable introduced into the equation is the one that has the highest correlation with output, which has been adjusted for the effect of the first variable. If the regression coefficient is statistically significant (using F-test), adding the next variable is implemented in the same way [41,42].

The presented methods were used to calculate models from all variables (Table 1). Additionally, OLS, Ridge, LASSO, and elastic net models have been reimplemented for the best subset of input variables computed from stepwise regression. All models calculated in the study were tested by leave-one-out cross validation (LOOCV). In this method, the data set is divided into two subsets: learning and testing (validation). In LOOCV, the test set is composed of a selected pair of data ($x_{i}$, $y_{i}$), and the number of tests is equal to the number of data n. During the cross-validation, $RMSE_{CV}$ error was calculated:$$RMSE_{CV}=\sqrt{\frac{1}{n}\overset{n}{\underset{i=1}{\sum }}{\left(y_{i}-{\hat{y}}_{-i}\right)}^{2}}\;$$
where n—number of patterns, $y_{-i}$—the output value of the model built in the *i*-th step of cross-validation based on a data set containing no testing pair ($x_{i}$, $y_{i}$), ${\hat{y}}_{i}$—the output value of the model built in the *i*-th step based on the full data set, and $RMSE_{CV}$—root mean square error of prediction.

## 3. Results

Table 2 summarizes the data regarding the participants and injuries characterization of Club Sport Marítimo in the 2020/2021 season. Of the 36 players participating in the study, 23 contracted at least one injury over the 2020/2021 season. Injured players missed an average of 14.3 days per injury. There were 0.9 injuries contracted by the number of participants (34 injuries/36 players) over the study period. Most injuries were classified as traumatic (52.9%). About 50% of the injuries were, according to their severity, moderate, since the athletes missed between 8 and 28 days of training and/or competition. Finally, four of the injuries counted were classified as recurrent.

Figure 1, Figure 2 and Figure 3 summarize the type, area, and specific location of injuries. The lower limbs were the body area most affected by injuries (85.2%). Sprains (35.2%) and muscle injuries (35.2%) were the most recurrent type of injuries throughout the study period, particularly in the ankles (29.4%), quadriceps (11.7%), and hamstrings (11.7%).

Table 3 presents the errors for each model and the sets of predictors calculated by the variable selection methods. The classical OLS regression model has the worst predictive ability, for which the error of RMSE = 18.57. Such a large error shows that the injury-prediction problem is complex and needs to be regularized by, among other things, using shrinkage regression. The use of shrinkage models (Ridge, LASSO, and elastic net) resulted in a sharp decrease in error and, thus, an improvement in the predictive ability of the model. The best model performing injury-prediction tasks for all predictors is the Ridge model, in which the RMSE error was 0.698. The optimal Ridge model was calculated for λ = 82.2. Optimizations of all shrinkage models are presented in Figure 4. The LASSO model for all predictors was not calculated because the algorithm does not work properly for such a configuration of the number of variables and patterns. Therefore, the following model used was the elastic net regression model. For elastic net regression, a very small prediction error was obtained (RMSE = 0.633), and the number of predictors was reduced due to the properties of this method. The result of the elastic net analysis was that the best set of input variables is the set of seven variables: x_1_—sectorial position 1, x_3_—sectorial position 3, x_7_—body height, x_12_—sit and reach, x_13_—*n* push-ups, x_15_—handgrip (l), and x_20_—V35 m.

The forward regression showed that the significant predictors are x_1_ – sectorial position 1, x_12_—sit and reach, x_13_—*n* push-ups, and x_15_—handgrip l). All the predictors determined by forward regression are contained in the set determined by elastic net regression. The model determined by forward regression generates an error of RMSE = 0.618. The predictors obtained using elastic net (E) and forward regression (F) were used in further predictive analysis. Both sets were used to recalculate the Ridge and LASSO models. The Ridge model with the set calculated by elastic net generates an error of RMSE = 0.592, and a very similar error was obtained for the Ridge model, with the set calculated by forward regression, with RMSE = 0.591. Both Ridge models with new sets of predictors show the best ability. LASSO models for enumerated sets of predictors showed worse predictive abilities than Ridge models. In the case of the best model, the model predicts the number of injury occurrences with an error of 0.59. This means that if a player has three injuries, the model would predict a value from the range of 2.41 to 3.59. The equations for the best models are presented in Table 4.

## 4. Discussion

This study aimed to analyze predictive modeling of injury risk based on players’ sectorial position, body composition variables (i.e., weight, height, TBW, FAT, and FFM), and selected physical fitness tests, which include sit-and-reach, push-ups, handgrip, CMJ, SJ, 5 m, 10 m, 35 m, and yoyo tests. 

This study considered multiple-input single-output regression-type models. It allowed us to select the best model to perform injury prediction tasks, considering all predictors. Previous work on predictive injury risk models is mostly based on classification learning models [31,43,44]. These models’ predictive accuracy ranged from 75% to 82.9% [30]. The present study did not use a categorical variable but rather a continuous variable. A similar solution was presented in another work, where a continuous variable was also placed in the output [45]. A direct comparison of the models’ predictive ability with those presented by other authors is complex because different quality criteria were used. 

The value of cross-validation error is important, but a more critical element of the analysis presented was the identification of significant predictors of injury risk. An important part of the analysis was the variable-selection methods, resulting in a very clear and simplified model structure. The simple structure of the model and the linear nature of the methods made it possible to interpret the impact of individual variables on injury risk. Data-selection mechanisms were also used by other authors who have also used LASSO [44].

According to the data collected for this study, a professional football team can experience 0.9 injuries for every player on the field. This number is noticeably lower than that reported in a study following the analysis of three sports seasons, averaging 1.5 injuries per player [4]. In reality, training load and competitive load—both internal and external—are variables that are related to muscle injuries and that change depending on the situation and level of competition. In this study, sprains and muscular injuries were the most common types of injuries in the lower limbs. The quadriceps and hamstrings were the next most afflicted muscles, followed by the ankles. These results are consistent with the previous findings in the literature [10,12,13,14,16]. In reality, the lower limbs are under more pressure in this activity because of the tactical–technical maneuvers needed, which justifies their increased risk of damage. Overload injuries were more common than traumatic injuries. A recent investigation also established the existence of such prevalence [4]. In contrast, a different article discovered that overload was the cause of two out of every three injuries in their study [12]. Since there is a strong link between training load and the likelihood of injury, it is imperative to emphasize the significance of appropriately structuring the training cycles according to the players’ attributes and physical condition. When individual training loads are measured using the right tools, this process happens more reliably and consistently. Coaches, players, and their technical-support personnel increasingly monitor and evaluate the sports load using a scientific method [46]. In reality, keeping an eye on the training process is essential for assessing the level of athlete weariness, which may help to lower the risk of injury. Soccer involves physical contact and high intensity. Therefore, injury-prevention procedures should take both overload and traumatic injuries into account. Each athlete missed 14.3 days of practice or competition after suffering an injury, on average. This finding differs from that seen in the literature, with players missing an average of seven to eight days owing to injury [4,8,12]. On the other hand, we draw the conclusion that more serious injuries result in a longer period of player absence. This demonstrates the necessity of strengthening all preventative and rehabilitation efforts, while taking into consideration the predictive variables of injury as well as more frequent medical checkups and physical testing. Some authors claim that muscle injuries in soccer are the most common [9,10], converging with our findings. The injury-recurrence rate in our study is consistent with the rates reported in the literature, which range from 8% to 22% [9,47,48]. According to earlier research, these percentage discrepancies may result from the resources available in the individual clinical departments as well as a particular club’s infrastructure and material-resource capabilities to respond quickly, in order to maximize the injury prevention and healing process.

Regarding the impact of selected predictors included in the models, first of all, for sectorial position, the defensive and forward sectors were the ones that presented a higher risk of injury. A previous study conducted across three consecutive seasons with 123 Chilean elite male football players also reported that the defensive and forward sectors were the ones that contracted more injuries over the study period [4]. Among 71 Spanish elite male players, forwards were the ones who presented the highest rates in both incidence and severity of injury [14]. Indeed, the literature has described that certain positions, such as fullbacks and forwards, have more demanding tasks both in-game and during training sessions, such as covering greater distances and running with higher intensity than their peers. Overall, fullbacks and forwards perform a total of 29–35 sprints, which is higher than other positions (approx. 17–23 sprints) [49], which may justify their higher injury rates (i.e., hamstring injuries) [50,51]. Therefore, managing training loads appropriately following the physical demands of different sectors and playing positions might be a helpful method to lower the risk of injury in football [52]. Sports agents and coaches should consider load exposure according to players’ position, particularly when designing training sessions [52]. Moreover, our results consolidate the need to consider the players’ position as a variable to be included in the definition of injury-risk programs. 

Another important predictor identified in our study was lower-limb flexibility. The sit-and-reach test is one of the physical fitness tests mostly used to predict the injury risk of elite football players across a sports season. In the literature, several studies have concluded that reduced flexibility in the lower limbs is related to the increased risk of injuries in elite football players [53,54,55,56]. Some studies report that it is essential to develop and introduce a standard battery assessment of flexibility in preseason tests, contributing to the awareness of the players’ profile [56,57]. The newest Guidelines for Exercise Testing and Prescription from the American College of Sports Medicine reported that maintaining good flexibility in all joints depends on many specific variables, including distensibility of the joint capsule and muscle viscosity, which facilitates movement and may prevent injuries [58]. However, we must acknowledge some limitations on the topic. First, it is not entirely understood if pre-activity stretching unequivocally reduces injuries associated with training load. Secondly, the most recent guidelines recommend direct measures of range of motion (i.e., goniometer and inclinometer) rather than indirect methods, such as sit-and-reach tests assessing flexibility. This means that most of the indirect measures that we most often use in various sports context are coming into disuse. It is recommended that direct measures of range of motion should be used more regularly. In general, the important focus will be that future studies continue to investigate this topic, so we can draw more reliable and valid conclusions regarding the relationship between flexibility and sports injuries.

According to our analyses, the push-up, handgrip, and 35 m linear sprint tests may be reliable predictors of injury risk among elite football players. Besides, height was also one of the variables significantly integrated into injury-prediction models in elite football players. Those variables can be related to each other, since they all end up influencing the players’ sports performance. In fact, the main value of this study is directed towards sports monitoring and injury prevention, as we analyzed the relationship between overall strength and height in elite soccer players as predictors of injury, and this is a topic on the rise. In the literature, we identified two studies conducted with youth footballers that have determined that injured players were significantly stronger, bigger, and more experienced than non-injured players [59,60]. This aspect becomes even more relevant when we talk about elite football players, since their demands are higher. The slightest physical differences can make all the difference in the outcome of individual action, dictating the outcome of crucial moments of games and seasons. We believe that these achievements can support future research on the topic to disentangle this complex net of variables that may affect the injury profile.

There are some limitations to this study that need to be acknowledged. The sample size and the fact that we only evaluated the elite players for 26 weeks across 42 weeks of the season are the main limitations of this study. The sample size is related to the number of patterns teaching predictive models. The greater the amount of recorded-injury information is, the better the material for calculating predictive models. Continued collection of learning patterns will improve the predictive ability of the models. Moreover, this is a cross-sectional study, which does not allow a cause–effect of the presented results. However, these results bring important and specific practical implications for those involved in the elite football context, mainly for the topics of injury prevention and training monitorization, since these are issues that are gaining significant attention in the sports business.

## 5. Conclusions

Addressing the need for further studies to identify risk factors for predicting injuries in elite football players, our approach opens a novel perspective on injury prevention and training monitorization, providing a methodology for evaluating and interpreting the complex relations between injury risk and players’ performance in elite football. Players’ sectorial position, body-composition variables, and physical fitness tests (sit-and-reach, push-up, handgrip, countermovement jump, squat jump, linear speed, and yoyo tests), were all important predictors that may be considered in the injury-risk prevention in elite football players. It would be an added value if future studies analyzed the influence of body-composition factors and physical fitness tests in elite football teams across different seasons.

## Figures and Tables

**Figure 1 jcm-11-04923-f001:**
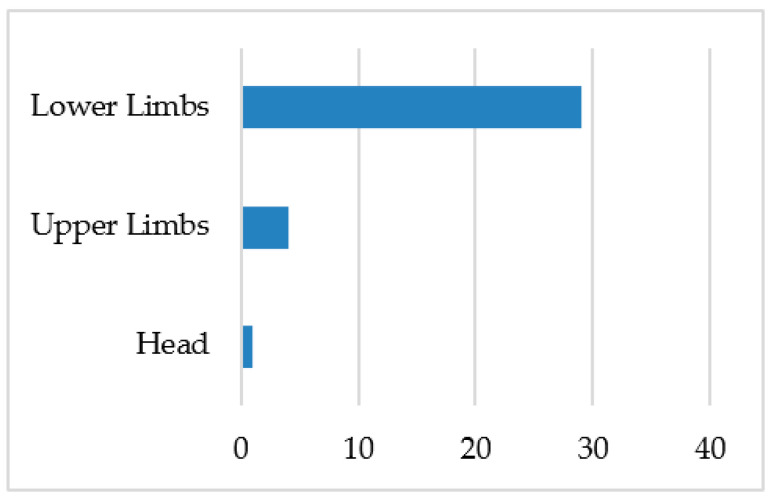
Injury frequency by zone (*n*).

**Figure 2 jcm-11-04923-f002:**
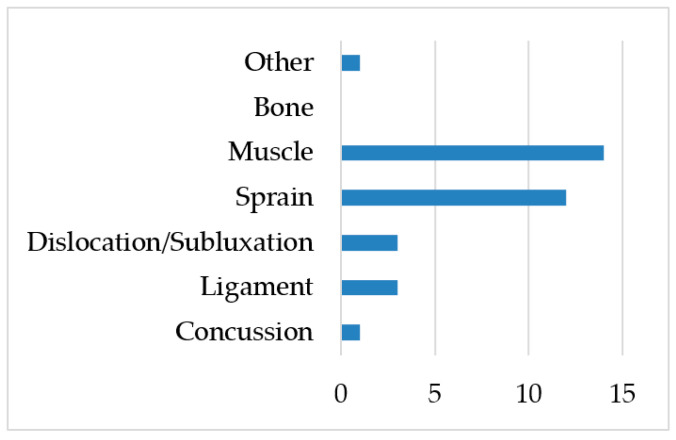
Injury frequency by type (*n*).

**Figure 3 jcm-11-04923-f003:**
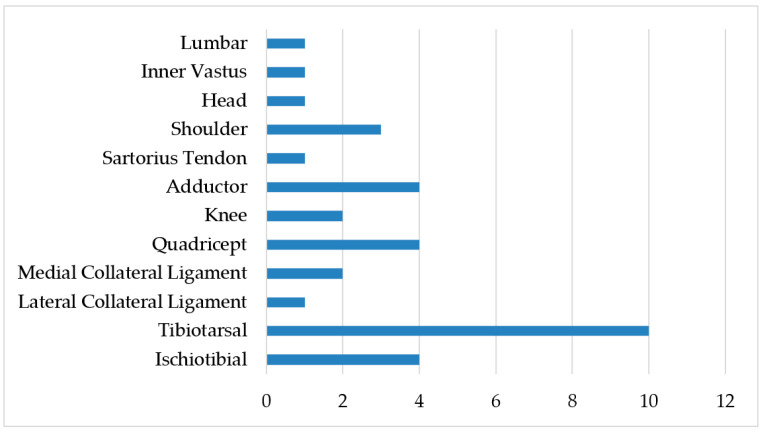
Injury frequency by specific location (*n*).

**Figure 4 jcm-11-04923-f004:**
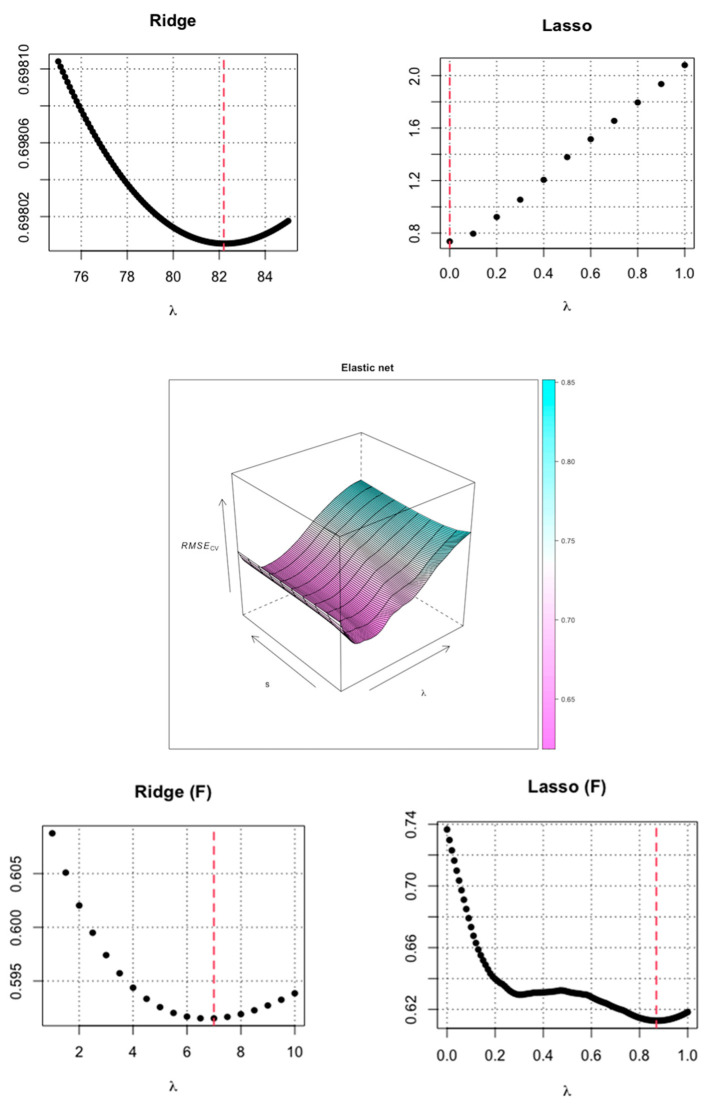
Optimization of predictive models (the red line indicates the optimal model).

**Table 1 jcm-11-04923-t001:** Description of the variables used to construct the predictive model (N = 24).

Variable	Description	M	sd
**x_1_**–x_3_	Sectorial Position *	-	-
x_4_	Age (y)	25.45	3.34
x_5_	Experience (y)	7.29	3.38
x_6_	Body mass (kg)	80.09	7.07
x_7_	Height (cm)	182.52	6.01
x_8_	TBW (L)	51.93	4.66
x_9_	BFM (kg)	8.2	2.41
x_10_	FFM (kg)	71.2	6.50
x_11_	Previous injury (*n*)	1.29	1.63
**x_12_**	Sit and reach (cm)	34.52	6.79
**x_13_**	Push-ups (*n*)	43.63	8.68
**x_14_**	Handgrip right (kg)	50.87	9.62
x_15_	Handgrip left (kg)	48.92	8.67
**x_16_**	CMJ height (cm)	40.14	4.58
x_17_	SJ height (cm)	39.64	4.26
x_18_	LS 5 m (s)	1.16	0.13
x_19_	LS 10 m (s)	1.88	0.16
x_20_	LS 35 m (s)	4.85	0.27
x_21_	Estimated VO2 max (L/kg/min)	50.82	3.98
x_22_	Yoyo (m)	1720	476
y	Injury frequency (*n*)	0.79	0.72

*—qualitative variable, M (mean value), sd (standard deviation), Me (median), TBW (total body water), BFM (body fat mass), FFM (fat free mass), CMJ (countermovement jump), SJ (squat jump), LS (linear speed), y (years), kg (kilograms), cm (centimeters), L (liters), *n* (number), s (speed), min (minutes), m (meters).

**Table 2 jcm-11-04923-t002:** Characterization of participants and injuries of CS Marítimo in the 2020/2021 season.

No. of Players	36
No. of Injured Players	23
Total Injuries	34
Average Days Missed Due to Injury	14.3
Injury per Player	0.9
*Injury Mechanism*	
Traumatic	18 (52.9%)
Overload	16 (47.1%)
*Injury Severity **	
Minimal (1–3 days)	4 (11.7%)
Mild (4–7 days)	7 (20.5%)
Moderate	17 (50%)
Severe (+28 days)	6 (17.6%)
*Injury Recurrence*	
Yes	4 (11.8%)
No	30 (88.2%)

* Number of days missed by a player due to a sports injury contracted in training or match.

**Table 3 jcm-11-04923-t003:** Predictive errors for calculated models.

Method	Predictors	RMSECV	Parameter
OLS	x_1_, x_2_, x_3_, …, x_23_	18.57	-
Ridge	x_1_, x_2_, x_3_, …, x_23_	0.698	*λ* = 82.2
LASSO	x_1_, x_2_, x_3_, …, x_23_	0.737	*s* = 0
Elastic net (EN)	x_1_, x_3_, x_7_, x_12_, x_13_, x_15_, x_20_	0.633	λ = 0.1, *s* = 0.22
Forward (F)	x_1_, x_12_, x_13_, x_15_	0.618	-
Ridge (EN)	x_1_, x_3_, x_7_, x_12_, x_13_, x_15_, x_20_	**0.592**	λ = 17.5
Ridge (F)	x_1_, x_12_, x_13_, x_15_	**0.591**	*λ =* 7
LASSO (EN)	x_1_, x_3_, x_7_, x_12_, x_13_, x_15_, x_20_	0.635	*s =* 0.55
LASSO (F)	x_1_, x_12_, x_13_, x_15_	0.613	*s =* 0.87

**Table 4 jcm-11-04923-t004:** Predictive errors for calculated models.

Method	Equation
Ridge (EN)	y = 0.01 + 0.10⊕x_1_ − 0.27⊕x_3_ + 0.01⊕x_7_ − 0.01⊕x_12_ − 0.01⊕x_13_ − 0.03⊕x_15_ − 0.45⊕x_20_
Ridge (F)	y = −0.28 + 0.35⊕x_1_ − 0.02⊕x_12_⊕−0.01x_13_ + 0.04⊕x_15_

## Data Availability

The data presented in this study are available upon request from the corresponding author.

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
