# Peer review of "Predictive Modeling of Injury Risk Based on Body Composition and Selected Physical Fitness Tests for Elite Football Players"

_jcm, 2022, doi:10.3390/jcm11164923_

Round 1
Reviewer 1 Report
This is a predictive study on the risk of developing injuries in professional football players. An important topic of research with potential impact on sports medicine practice and training methods. I find this study strongly focusing on statistical methods (which is OK) but poorly describing the clinical components to better interpret the findings. A more balance reporting between statistics and clinical components are necessary for the readership of JCM. As an academic clinician I would like to see more, on Results section, on the injuries’ characteristics not just its mean/median number for modelling: i.e, the location, type (e.g., traumatic, overuse, muscular, ligamentous), time lost in competition/practice, etc. This will help me and others critically appraise how “real context” is your predictive model. Also, sample size seems very small for machine learning methods hence difficult to appraise the performance of the prediction and how reliable is it (1).
1 – Riley R D, Ensor J, Snell K I E, Harrell F E, Martin G P, Reitsma J B et al. Calculating the sample size required for developing a clinical prediction modelBMJ 2020; 368 :m441 doi:10.1136/bmj.m441
Specific comments:
Abstract
Line 38: This sentence seems like a conclusion, nevertheless I suspect many readers won’t understand this point of view based on the rest of the abstract.
Line 41: This is an irrelevant sentence if not is explained why. It is not easily inferred from past sentences.
Keywords: You may diverse your key words. Many are already in the title.
Introduction
Line 62: You may add this reference, Van Eetvelde H, Mendonça LD, Ley C, Seil R, Tischer T. Machine learning methods in sport injury prediction and prevention: a systematic review. J Exp Orthop. 2021;8(1):27. Published 2021 Apr 14. doi:10.1186/s40634-021-00346-x
Line 63: “showed” instead of “show”
Lines 64–66: This study did not investigate the mechanisms for lowering of the injury rate. This is speculative or a hypothesis at best. Please rephrase so that readers realize that the causes are not yet known but that there are potential explanations (still to be studied).
Line 84: I would say that “neuromuscular” belongs to the umbrella term “physiological”. Perhaps you mean physical performance tests?
Line 96: This last part of the sentence is redundant (considering the last sentences).
I suggest you remove “to analyze prediction modeling of injury risk”.
Methods
Line 102: If you are reporting median why not the quartiles as well? Or you report x ± sd for normally distributed variables and median and Q2,Q3 for non-normally distributed variables. That would be more rigorous.
Table 1: List abbreviations at the end of the table, please. I suggest that you may add “estimated” before VO2max so that readers can easily identified that it was not a direct measurement.
Lines 170–175: Please emphasize that VO2 was an estimative based on performance of yoyo test, not directly measured. Or was it? With a portable device? Which brand?
Line 198: I there a threshold for parameter λ?
Line 214: report the threshold for being considered “significant”
Line 219: Please inform readers how to interpret RMSE values.
Results
As an academic clinician I would like to see more on the injuries characteristics not just its mean/median number: the location, type (e.g., traumatic, overuse), time lost in competition/practice, etc. This will help me and other critically appraise how “real context” is your predictive model.
Line 238: This sentence would be better suited in the Methods section.
Line 250: OK, but how good are your best models to predict injuries, i.e., the performance or prediction ability of the model?
Discussion
You suddenly changed the style of the citations in text.
Line 329: How many participants would have this study needed to be more robust statistically?
Line 332: When you state “generalization” do you mean cause-effect?
Conclusions
Please reword, providing a conclusion of your findings (facts). Not some vague potential use of the methods used.
Reviewer 2 Report
Thank you very much for the peer review request. This study aimed to analyze predictive models of injury risk in soccer players, and I found the entire manuscript interesting.
Please find below my comments and refer to them as necessary.
Line 61-61: Although you are focusing on elite football, how widely is Machine Learning used in sports? Is Machine learning being used in other sports or non-elite football?
Does this contradict the explanation in Line 72-81. Or do these focus on different points?
Line 64: What does “These results” show? If “An investigation…”, I recommend to revise to “this result”
Line 92-93: “the complex patterns that underlie injury prediction”. What does this indicate? Please be specific.
Line 95-97: Which is the novelty of this study to analyze prediction modeling of injury risk or to target Portuguese athletes? If it is the latter, some evidence in global study and reasons for targeting Portuguese athletes will be needed.
Line 100: It is very important to state whether these players belong to the same team or different teams within the same league. If they are on the same team, any bias should be taken into account.
Table 1: Why is the subject number 24, not 36.
Table 1: For some Ms, please align the decimal points (X5,6,10,11).
Line 127: Are these physical fitness assessments performed during a day. Did you split the program into multiple days to account for fatigue? Have intertest effects from each test been eliminated?
Line 269: While most previous studies have used methods based on classification learning models, this study used continuous variables, referring to Ruby et al. Please explain why using continuous variables can improve prediction accuracy (I am not an expert in this area, so if there is no need to discuss it, it is not necessary).
Line 287: “fullback”. Is this a generalizable term? Is it different from defender?
Line 311: This explanation is most importance in this part. I think there are many reports regarding stretching prior to activities and injuries. Do you mean that while there are reports that stretching prior to activities is effective in preventing injuries, there are also scattered reports that criticize it?
Line 312-313: I think there is a lack of explanation as to why this is a problem. Please add an explanation.
Line 318: Those variables should be related to each other. Why?
Line 318-319: “we are unaware of any study that has analyzed the relationship between elite football players’ general strength”. A literature review should be conducted.
Round 2
Reviewer 1 Report
In general, I’m satisfied with authors amendments and more clinical information provided. I believe the quality of the reporting and article had significantly improved. A few suggestions and some doubts persist though.
Suggestions/amendements:
· Line 37 – I apologize not having mentioned before, in my first review, but for clarity, I suggest: “…seat-and-reach performance, 1-minute number of push-ups, handgrip strength, and 35m linear speed.”
· Table 2 information is very limited hence unnecessary to be in a table. Perhaps Tables 2 and 3 are better fused?
· Last paragraph of the Discussion section – Explanation why sample size is a limitation and how it can be overcome in future studies should be provided.
Questions:
· Considering the title of Tables 1 and 2, the 34 football players were from only one club. I find it hard to believe that a team can manage 34 professional football players. Can you comment on that?
· I find the explanation of the RMSE interpretation more understandable but still very confusing considering the example of 3 injuries estimative when your data on injuries are far from that. Can you please provide a better example, more closely to the data you present? For example, is it possible to use the graphs that are displayed in Figure 4 to illustrate your point? That could help a lot. Thank you.
